# Modification of Physiochemical and Techno-Functional Properties of Stink Bean (*Parkia speciosa*) by Germination and Hydrothermal Cooking Treatment

**DOI:** 10.3390/foods12244480

**Published:** 2023-12-14

**Authors:** Seema Vijay Medhe, Aurawan Kringkasemsee Kettawan, Manoj Tukaram Kamble, Nuntawat Monboonpitak, Kim D. Thompson, Aikkarach Kettawan, Nopadon Pirarat

**Affiliations:** 1Department of Food Chemistry, Institute of Nutrition, Mahidol University, Salaya, Phutthamonthon, Nakhon Pathom 73170, Thailand; seemamedhe@gmail.com (S.V.M.); aurawan.kri@mahidol.ac.th (A.K.K.); nuntawat.mon@mahidol.ac.th (N.M.); 2Wildlife, Exotic and Aquatic Animal Pathology Research Unit, Department of Pathology, Faculty of Veterinary Science, Chulalongkorn University, Bangkok 10330, Thailand; maav.manya@gmail.com; 3Moredun Research Institute, Penicuik EH26 0PZ, UK; kim.thompson@moredun.ac.uk

**Keywords:** germination, hydrothermal cooking, stink beans, physiochemical properties, techno-functional properties, FTIR, microstructural, morphological, food security

## Abstract

Stink bean, *Parkia speciosa*, is recognized as a significantly underutilized legume with versatile utility and diverse benefits. However, information on the impact of different processing methods, such as germination and hydrothermal cooking, is scarce on stink beans (SBs). Therefore, the current research aimed to explore the efficacy of germination (G) and hydrothermal cooking (HTC) on the physiochemical properties, proximate composition, techno-functional properties, and antioxidant potential of SB flour. Furthermore, Fourier transform infrared spectroscopy (FTIR) and field emission scanning electron microscopy (FESEM) were employed to assess structural and morphological changes. The results revealed that the physiochemical properties of SB were significantly enhanced through processing, with more pronounced improvements observed during germination. Additionally, SBG exhibited a significantly higher protein content and lower fat content compared to SBHTC and stink bean raw (SBR). Moreover, techno-functional properties such as color intensity, least gelation concentration, and pasting properties were significantly improved in SBG compared to SBHTC and SBR. FTIR analysis of SBG and SBHTC indicated structural modifications in the lipid, protein, and carbohydrate molecules. FESEM examination revealed morphological changes in SBG and SBHTC when compared to SBR. Importantly, SBG exhibited higher antioxidant activity and total phenolic content in comparison to SBHTC and SBR. Therefore, processed SB flour can be incorporated and utilized in product development, highlighting its potential as a plant-based protein source for protein-rich breakfast bars and cookies.

## 1. Introduction

Pulses are extensively consumed for human nutrition in developing countries due to their cost-effective and nutrient-dense source of protein and carbohydrates [1]. Furthermore, pulses are increasingly used in dietetic formulations to treat and prevent diabetes, cardiovascular disease, colon cancer, and hypercholesterolemia [2]. Protein-energy malnutrition is rampant in developing and underdeveloped countries due to the limited availability of protein sources [3]. Plant proteins are used as food ingredients due to their functional properties, such as emulsification, solubility, foaming, water or oil binding, and gelling [4]. Nonetheless, these properties influence food texture and organoleptic characteristics, essential in manufacturing products like confectioneries, beverages, dressings, and meat products [5]. Moreover, the size, shape, density, porosity, and coefficient of friction of pulse seeds hold significance in the design of equipment employed in the harvesting, transporting, cleaning, separating, packaging, storing, and processing of various foods [6].

The food industry has witnessed a consistent increase in demand for innovative ingredients obtained from natural sources. Consequently, this demand has attracted the interest of researchers in the ingredients obtained from agricultural products. While numerous varieties of legumes are found worldwide, the emphasis on the utilization of many of these legumes is primarily placed on kidney beans (*Phaseolus vulgaris*), soybeans (*Glycine max*), and cowpeas (*Vigna unguiculata*) [7,8]. Hence, there is a substantial gap in the exploration of underutilized legumes, which are predominantly restricted to localized regions around the world [8]. The stink beans, or petai (*Parkia speciosa*), are green seeds packed into flat, wavy, and elongated pods. They are famous in several Asian countries, including India, Malaysia, Thailand, and along the Indo-Pacific coast [9]. The flowers, seeds, and pods of *P. speciosa* plants are edible parts of the plant [10]. Furthermore, commercial products derived from this species are available in markets year-round and known by various regional names, such as sataw, chou-dou, smelly beans, sotor, petai, and u’pang [10]. The petai seed is rich in bioactive plant sterols like β-sitosterol and stigmasterol, phenolics, flavonoids, and various peptides [11,12,13]. Furthermore, it exhibits a range of potent biological activities, including antioxidant [14], antibacterial [15], antitumor [16], as well as anti-hypertensive effects [13].

Basic household methods like cooking and germination have emerged as a budget-friendly strategy for addressing anti-nutritional factors in beans, enhancing their sensory appeal [5]. Germination, a controllable and straightforward method, holds promise in augmenting the nutritional profile of grains. Under controlled humidity, grains germinate after a specific soaking period in water. This process leads to the breakdown of key nutrients like proteins, starch, lipids, and fibers, accompanied by increased hydrolytic enzyme activity. This transformation, in turn, reduces the cooking time of hard-to-cook seeds and enhances various functional aspects [17]. On the other hand, cooking triggers starch gelatinization, protein denaturation, and the solubilization of polysaccharides. The increasing desire for innovative raw materials in the functional food industry has driven a consistent market demand for beans processed using techniques such as germination and hydrothermal cooking. These techniques have proved effective in improving both the functional and nutritional qualities of beans [1,18]. Based on our current knowledge, there is a notable absence of published data on the physiochemical characteristics of raw, germinated, and hydrothermally cooked stink bean flour. The aim of this study is to examine how the functional and physiochemical properties of stink bean flour is affected by processes such as germination and cooking. Since various factors can influence properties like emulsion, foaming, thermal characteristics, and morphology, we explored different parameters to understand the diverse effects processing of stink beans can have. This knowledge is important for considering stink bean flour as a viable and economical raw material in the development of food products.

## 2. Materials and Methods

### 2.1. Experimental Treatment Preparation

#### 2.1.1. Raw Stink Bean (SBR)

Stink bean fresh pods, harvested from September to October, were purchased from the local marketplace in the Salaya sub-district, Nakhon Pathom, Thailand. In the laboratory, the pods were inspected for visible physical or insect infestation, and any defective pods were discarded. Subsequently, the seeds were manually dehulled from the pods and inspected for insect damage. Importantly, damaged seeds were discarded, and only whole, healthy seeds were lyophilized for 48 h. They were then milled to obtain flour with a particle size of 500 µm, according to the USA standard testing sieve No. 35, using a Philip blender (HR2118/02, Batam City, Riau Islands, Indonesia). The flour was kept at −20 °C in airtight glass containers for subsequent analysis [18].

#### 2.1.2. Germinated Stink Bean (SBG)

Stink beans were wrapped in a clean, disinfected, damp cotton towel, sealed in an airtight container, and stored in the dark at room temperature for 5 days. The damp cotton towel was replaced every 12 h with a fresh, sterilized towel. After 5 days of stink bean germination, the radicles had grown to be 3–5 cm long. The germinated seeds were freeze-dried for 48 h at −80 °C, and then finely ground to obtain flour with a particle size of 500 µm, using a Philips blender (HR2118/02, Batam City, Riau Islands, Indonesia). The flour was kept at −20 °C in an airtight glass container for subsequent analysis.

#### 2.1.3. Stink Bean Hydrothermally Cooked (SBHTC)

Stink beans were cooked at a ratio of 1:10 in distilled water for their optimum cooking time. After cooking, the seeds were patted dry using tissue paper to eliminate any excess surface moisture before being freeze-dried for 48 h at −80 °C. The freeze-dried seeds underwent pulverization using a Philip blender (HR2118/02, Batam City, Riau Islands, Indonesia) and passed through a sieve conforming to the USA standard (with a mesh size of No. 35) to obtain the seed powder. The powder was kept at −20 °C in an airtight glass container for further investigation.

### 2.2. Physiochemical Characteristics of Raw (R), Germinated (G) and Hydrothermally Cooked (HTC) Stink Beans (SB)

#### 2.2.1. Length, Width, and Thickness

The length, width, and thickness of 100 randomly selected seeds of SBR, SBG, and SBHTC were measured using a Vernier caliper with an accuracy of 0.05 mm in triplicate. For each seed, measurements of length, width, and thickness were divided into 4 equal lots, and 25 samples were randomly selected from each lot to obtain a total of 100 samples for the experiment. Consequently, measurements of all sizes and shapes were replicated one hundred times. The seed’s three principal axial dimensions, namely, length (L), width (W), and thickness (T), were measured using a manual Vernier caliper (Aerospace, Changsha, Hunan, China) with a 0.05 mm accuracy. Seed size was considered an important parameter in processing bulk samples.

#### 2.2.2. Equivalent Diameter

Seeds equivalent diameter (Dm) was evaluated using the formula below:Dm = (LWT)^1/3^,(1)
where L = length, W = width, and T = thickness.

#### 2.2.3. Sphericity

The calculation of sphericity (ϕ) was performed using the formula below:ϕ = [(LWT)^1/3^/L] × 100,(2)
where L = length, W = width, and T = thickness.

#### 2.2.4. Aspect ratio

The seed’s aspect ratio (Ra) was determined using the following formula:Ra = W/L,(3)
where W = width and L = length.

#### 2.2.5. Seed Volume

The seed volume (V) was determined using the formula below:V = πB^2^L^2^/6(2L − 3),(4)
where B = (WT)^1/2^, L = length, W = width, and T = thickness.

#### 2.2.6. Surface Area

The surface area (A) was determined using the following formula:A = πBL^2^/2L − B,(5)
where B = (WT)^1/2^ and L = length.

#### 2.2.7. Weight of Seed

The randomly selected 100 seeds of SBR, SBG, and SBHTC were measured utilizing a Mettler Toledo scale (XP2003SDR, Greifensee, Switzerland) with an accuracy of 0.001 mg in triplicate.

#### 2.2.8. Seed Volume

One hundred SBR seeds were placed in a 1000-mL measuring cylinder with 500 mL of distilled water. The difference in volume was recorded and divided by 100 to calculate the volume per seed [19]. Similarly, the volumes of 100 SBG and SBHTC seeds were also determined using this method.

### 2.3. Field Emission—Scanning Electron Microscopic (FE-SEM)

Before applying a platinum coating, lyophilized samples were fixed to an SEM specimen stub using double-sided tape [1,20]. The coating process was carried out using a Polaron SC 515 Sputter Coater (Fissions Instruments, VG Microteach, Sussex, UK). Subsequently, the specimens were imaged using a Leo Supra 50VP Field Emission Scanning Electron Microscope, equipped with an Oxford INCA 400 energy dispersive X-ray microanalysis system (Oxford Instruments Analytical, Bucks, UK).

### 2.4. Proximate Composition

The moisture, crude fat, ash, crude protein, total dietary fiber, and soluble and insoluble dietary fiber contents of the SBR, SBG, and SBHTC flour were determined following the official AOAC methods [21]. The carbohydrate content in both raw and processed stink bean and stink bean flour was calculated using the difference method. The energy (kcal/100 g) was assessed using the below-given equation [22]:Energy = (% Carbohydrate × 4 kcal) + (% Protein × 4 kcal) + (% Fat × 9 kcal).(6)

### 2.5. Techno-Functional Characteristics

#### 2.5.1. Hunter Color Measurement

A color spectrometer (ColorFlex EZ, HunterLab, Reston, VA, USA) was employed to analyze the color of seeds and flour across various scales (CIE L*, a*, and b*). The calorimeter was calibrated using standard white and black plates. In this context, L* denotes brightness (ranging from 0 for black to 100 for white), a* indicates reddish (+a*) and greenish (−a*) colors, and b* signifies yellowish (+b*) and blueish (−b*) colors. The measurements were conducted under consistent room temperature and lighting conditions. Additionally, each flour and bean sample underwent duplicate repetitions to ensure precision.

#### 2.5.2. Least gelation Concentration (LGC)

The determination of LGC was carried out in accordance with the method of Medhe et al. [18]. For each type of bean flour, dispersions were prepared using 2, 4, 6, 8, 10, 12, and 14 g of flour/100 mL water. These dispersions were prepared in test tubes, each containing 3 mL of distilled water, and the subjected to heating in water bath for one hour at 95–100 °C. After the heating process, the dispersions were allowed to cool to 4 °C. To determine the least gelation concentration, a visual assessment was conducted by observing whether any drops from the emulsion separated and rose to the top of inverted tubes. The results were then quantified and recorded accordingly.

#### 2.5.3. Emulsion Properties

The evaluation of emulsion capacity (EC) and stability (ES) was conducted following the method of Medhe et al. [18]. Briefly, a 5% (*w*/*v*) flour suspension of 100 mL was homogenized using an IKA T25 digital Ultra-Turrax homogenizer at a speed of 24,000 rpm for 2 min. Subsequently, 100 mL of soybean oil (with a density of 0.912 g/mL) was added to each sample and homogenized for an additional 2 min. To determine emulsifying activity, 10 mL of resulting emulsion were centrifuged at 2500× *g* for 5 min, and the volume of the emulsified layer was measured. The emulsifying activity was quantified as a percentage, reflecting the emulsified layer’s volume relative to the entire layer in the centrifuge tube. On the other hand, emulsion stability was assessed by subjecting the prepared emulsions to heat treatment at 80 °C for 30 min. Subsequently, the emulsion was allowed to cool at room temperature and centrifuged at 1200× *g* for 5 min. The stability of the emulsion was determined as a percentage, representing the remaining emulsified layer’s volume in relation to the original emulsion volume.

#### 2.5.4. Swelling Capacity (SC)

Twenty grams of seeds, or cotyledon, were measured in triplicates and soaked in distilled water overnight [1]. The volume of soaked seeds, or cotyledons, was measured using a measuring cylinder.
SC (mL/seed or cotyledon) = (V_2_ − V_1_)/N,(7)
where V_1_ = volume of seeds or cotyledons before soaking; V_2_ = volume of soaked seeds or cotyledons; and N = number of seeds or cotyledons.
Swelling index (SI) = swelling capacity of seed or cotyledon/Volume of one seed or cotyledon(8)

#### 2.5.5. Hydration Capacity

Twenty grams of seeds or cotyledons were measured and placed in a measuring jar containing 100 mL of distilled water in triplicate and left it for 24 h at room temperature (28 ± 2 °C). Afterwards, the water was drained and seeds or cotyledons were blotted to remove any adhering water and then weighed [19].
Hydration capacity (g/seed) = (W_2_ − W_1_)/N,(9)
where W_1_ = weight of seeds/cotyledons before soaking; W_2_ = weight of soaked seeds/cotyledons after soaking; and N = number of seeds/cotyledons.
Hydration index = hydration capacity per seed/weight of one seed(10)

#### 2.5.6. Pasting Properties

The pasting characteristics of the bean flour were assessed using the Rapid Visco Analyzer (RVA-4, Newport Scientific Pvt., Ltd., Warriewood, NSW, Australia). Flour suspensions, with a total weight of 29 g, were prepared to record the viscosity profiles of the flour. These suspensions underwent a heating process from 50 to 95 °C at a rate of 6 °C per min (after a 1 min equilibration period at 50 °C), were then maintained at 95 °C for 5 min, subsequently cooled from 95 to 50 °C at the rate of 6 °C per min, and finally held at 50 °C for 2 min [23].

#### 2.5.7. Water Absorption and Oil Absorption Capacity

Fifty milligram samples of flour were placed into a pre-weighed Eppendorf tube [1]. Subsequently, 1 mL of distilled water was added for one set of samples, while for another set to assess oil binding, 1 mL of soyabean oil was added. The samples were then vortexed and allowed to stand for 30 min at controlled temperatures of 25 ± 2 °C. Following the incubation, the sample was centrifugated at 2500× *g* for 25 min. Excess water or oil was eliminated by inverting the tubes over absorbent paper and allowing samples to drain. The weights of the retained water and bound oil samples were determined by calculating the difference in weight before and after this process.

### 2.6. Fourier-Transform Infrared Spectroscopy (FTIR) Analysis

An FTIR spectrometer (Nicolet Summit Pro, Thermo Scientific, Waltham, MA, USA) was used to obtain the FTIR spectra of stink bean flour in their raw, germinated, and hydrothermally cooked forms [1]. According to the protocol, the flour of stink bean and KBr were combined in a proportion by weight of 1:100. The resulting combination was then crushed and compacted into a pellet. The measurements were carried out using a spectral resolution of 4 cm^−1^, and the range of measurements was 500–4000 cm^−1^. Sixty-four scans in all were gathered and averaged. It should be emphasized that the FTIR analysis was conducted three times.

### 2.7. Gelatinization Characterization

Differential scanning calorimetry (DSC) was employed to assess the gelatinization characteristics of stink beans [18]. Briefly, 20 mg of bean flour was placed into a hermetically sealed aluminum pan, along with 20 µL of water. Subsequently, the sealed pan was equilibrated for 1 h at 20 °C. The heating of the sample was conducted within an empty pan, ranging from 20 to 115 °C, at a heating rate of 2.5 °C per min, all while maintaining a nitrogen (N_2_) atmosphere at a flow rate of 50 mL per min within the DSC chamber.

### 2.8. Antioxidant Properties

#### 2.8.1. Total Polyphenol Content (TPC)

The quantification of TPC was conducted employing the Folin–Ciocalteu reagent method [24]. Briefly, a 0.2 g sample of flour was extracted with 4 mL 80% methanol at 30 °C in a water bath with shaking for 2 h. Following this, centrifugation at 2000× *g* for 10 min was performed. Subsequently, in a test tube, 500 µL of distilled water and 10 µL of extract were combined, and 50 µL of Folin–Ciocalteu reagent (Sigma Aldrich, St. Louis, MO, USA) was promptly added to initiate the reaction. Following a 3-min incubation, 245 µL of distilled water and 200 µL of sodium carbonate (Na_2_CO_3_) solution (20 g/L (*w*/*v*)) was added to the reaction mixture. A spectrophotometer was utilized to measure the absorbance of the test solution and standard (Gallic acid, 10–80 μg/mL). The TPC quantification was represented as mg/100 g of gallic acid equivalent using the following computation, which is based on the standard curve equation: y = 0.0063x + 0.0245, R^2^ = 0.9993 (where x is the standard of total phenolic content and y is the absorbance at 750 nm).

#### 2.8.2. Assessment of Antioxidant Activity by DPPH Assay

The assessment of antioxidant capacity was carried out utilizing the 1,1-diphenyl-2-picrylhydrazyl (DPPH) radical method [25]. Briefly, a 2 mL solution of methanolic extract was mixed with 2 mL of a DPPH solution (0.15 mM, 95% methanol) and allowed to incubate at room temperature for 30 min without exposure to light. Following this, the absorbance of both stink bean samples and a reference substance, Trolox, was measured at a wavelength of 517 nm. The flour’s ability to neutralize free radicals was quantified as mM Trolox equivalent (TE) per gram of sample. This analytical procedure was conducted in triplicate.

#### 2.8.3. Oxygen Radical Absorbance Capacity (ORAC) Assay

The ORAC analysis was performed following the method of Alberto et al. [25]. In brief, the sample (20 µL) was placed into a 96-well microplate. Subsequently, the microplate was tightly sealed using parafilm and placed in a fluorometer (FLOU Star OPTIMA Microplate Reader, PerkinElmer, Buckinghamshire, UK) for an incubation period of 30 min at 38 °C. After the initial incubation, an additional 10 min incubation was performed following the removal of cover. Thereafter, 200 µL of a fluorescent solution and 20 µL of a 3.2 mM 2,2′-Azobis(2-amidinopropane) dihydrochloride (AAPH) solution were added to the wells. The change in fluorescence, as detected at 485 nm (excitation) and 520 nm (emission), was utilized for kinetic evaluation. The linear regression curve from the standard (Trolox) or the sample, in combination with the area under the fluorescence decay curve, was used to calculate the final values of ORAC. These values were expressed as micromoles of Trolox equivalent per gram of the sample (µM TE/g).

### 2.9. Statistical Analysis

The physiochemical parameters were assessed in a sample size of 100 replicates, while the proximate composition and techno-functional properties were determined based on three replicates each. Additionally, the antioxidant activity and total polyphenol content were evaluated in triplicate, and outcomes are expressed as mean ± standard error. Statistical analysis was conducted utilizing SPSS software (version 28, SPSS Inc., Chicago, IL, USA), and Tukey’s HSD multiple range test was employed to determine significant differences (*p* < 0.05) among the mean values.

## 3. Results and Discussion

### 3.1. Physiochemical Characteristics

SBG (27.15 mm) exhibited a significantly greater length compared to SBR (24.36 mm) and SBHTC (21.41 mm) (Table 1). The germination process can lead to an increase in seed length due to enhanced water absorption during processing. Similar results were found for the width, thickness, and diameter of SBG. These findings align with a previous study on dry beans [26].

The greatest increase in seed width and thickness was observed in germinated seeds. Germination enhances the permeability of cell walls, diminishes the density of the intracellular environment, and alters the macrostructures of proteins, potentially facilitating their absorption of water from the surrounding matrix more effectively [27]. The processing enhances porosity and augments the rate of water absorption. The volume, surface area, and weight of SBG exhibited statistically significant elevation compared to SBR and SBHTC. Consistent with previous studies, it has been observed that both the weight and volume of beans increased following the germination process [1,28]. The shape of food materials is typically characterized by their sphericity and aspect ratio. In the present study, the SBR found significantly greater sphericity and aspect ratio compared to the SBG, which is in accordance with previous studies [1,29]. It was noted that the dimensional attributes such as length, width, diameter, and thickness were greater in processed stink beans compared to their raw seeds.

### 3.2. Proximate Composition

The moisture content in SBHTC (3.78 g/100 g) was significantly lower compared to that in SBR (4.49 g/100 g) and SBG (4.38 g/100 g) (Table 2), which may be attributed to the process of lyophilization [1].

The protein content was significantly higher in SBG (35.61 g/100 g) than in SBHTC (33.68 g/100 g) and SBR (31.40 g/100 g). Similarly, previous studies reported that the protein content of mung beans increased significantly after germination [30,31]. Nevertheless, the germination process initiates with protein hydrolysis and synthesis, which leads to an increase in protein content [32]. This enzymatic activity mainly occurs due to the presence of protein bodies within the testa and axis parts of seeds, facilitating the growth of new tissues and consequently increasing the water-soluble proteins [33]. Furthermore, the upsurge in protein content after germination can be attributed to the generation of plant hormones by the seed and/or their release during germination, activating and releasing functional proteins such as amylases, lipases, and proteases [34]. The higher protein content observed in SBHTC compared to SBR may be attributed to the elimination of thermolabile protein inhibitors and the denaturation of proteins due to heat—a process that facilitates the unfolding of globulin structures, rendering them more accessible. These findings align with prior research in the literature [33,35].

The fat content in SBG (20.29 g/100 g) exhibited a significant reduction when compared to SBHTC (28.73 g/100 g) and SBR (23.18 g/100 g). Similarly, previous research has indicated that the process of germination resulted in a reduction in the crude fat content of various legumes, including mung beans [36], cowpea, green gram, lentils, and chickpeas [31], as well as buckwheat and hemp seed [37]. The decline in crude fat percentage after germination can be attributed, in part, to the metabolism of stored fats, which are converted into elements of plant cell walls and other plant materials to support sprout growth [37]. Additionally, this reduction may be linked to the catabolic process that occurs during germination, utilizing stored fats.

The SBHTC (29.90 g/100 g) found significantly higher levels of total dietary fiber in comparison to SBR (24.96 g/100 g) and SBG (23.02 g/100 g). Similarly, previous studies have reported increased levels of total dietary fiber after cooking treatment in beans and chickpeas [38], as well as in eight varieties of *Phaseolus vulgaris* beans and two varieties of *Phaseolus coccineus* [39]. The increase in total dietary fiber content in SBHTC is due to the presence of resistant starch or the generation of protein–tannin composites [39]. Furthermore, SBHTC had significantly higher contents of soluble and insoluble dietary fiber than SBR and SBG. These findings are consistent with prior research that observed increased soluble dietary fiber (SDF) content in cooked beans and chickpeas compared to their raw counterparts [40,41]. Furthermore, the hydrothermal treatment applied to the beans and chickpeas led to an elevation in the amount of insoluble dietary fiber [38]. The process of hydrothermal treatment seems to induce the structural integrity of cell walls and storage of polysaccharides within pulses, potentially impacting tissue histology integrity and disrupting protein–carbohydrate interactions. As a result, this alteration leads to a decrease in the solubility of dietary fiber.

The decrease in ash content observed in SBHTC can be attributed to the leaching of both macro and micronutrients occurring throughout the soaking and cooking processes [18]. Additionally, mineral migration into the water during cooking can also contribute to the reduction of ash content [35,38]. The reduced carbohydrate content in SBG (37.97 g/100 g) in comparison to SBR (39.16 g/100 g) can be ascribed to the metabolic conversion of carbohydrates serving as substrates for energy generation during the germination [33], which is consistent with previous studies on germinated chickpea seeds [42] and desi kabuli chana [43]. Furthermore, SBHTC (527.53 kcal) was found to possess the highest energy value in comparison to the SBR (490.86 kcal) and SBG (476.93 kcal), consistent with a previous study on hydrothermally cooked Nitta bean [1].

### 3.3. Techno-Functional Properties

#### 3.3.1. Hunter Color

Table 3 displays the values of hunter color for SBR, SBG, and SBHTC. SBRF (41.33) exhibited a significantly lower degree of lightness in comparison to SBGF (44.48) and SBHTCF (44.36).

Conversely, SBG (45.08) seeds displayed substantially lower lightness levels than both SBR (49.33) and SBHTC (44.59) seeds. The L values suggest that SBRF possesses greater degrees of darkness compared to SBGF and SBHTCF, which are in accordance with the prior study which reported decreased lightness and increased yellowness in germinated brown rice [44]. Germination induces enzymatic hydrolysis, facilitating the release of amino acids. Nevertheless, the interaction between amino acids and soluble sugars triggers the Millard reaction, which may underlie the color alteration during germination [17]. The reduction in lightness during the germination of flour was notably more pronounced, a phenomenon that can be ascribed to variations in the amylose content [17].

The hunter color ‘a*’ values of SBHTC (0.20) were found to exhibit a higher degree of whiteness compared to SBRF (−4.92) and SBGF (−4.35). Similarly, hunter color ‘a*’ values of SBG (−7.33) indicated significantly (*p* < 0.05) higher greenness than the SBR (−9.21) seeds. Additionally, the hunter color ‘b*’ values of SBHTCF (20.15) manifested significantly (*p* < 0.05) increased levels of yellowness in contrast to SBRF (19.45) and SBGF (19.43), suggesting that germination and hydrothermal treatment amplified the yellowness of stink bean flour. Notably, the seeds of SBR, SBG, and SBHTC displayed significantly (*p* < 0.05) different levels of yellowness. Furthermore, SBGF (19.91) exhibited notably lower levels of color intensity when compared to SBRF (20.07) and SBHTCF (20.15). In contrast, SBR (36.04) displayed significantly (*p* < 0.05) highest levels of color intensity in comparison to SBG (32.38) and SBHTC (28.09) seeds. A similar hue angle pattern was observed for both the flour and seeds of stink beans.

#### 3.3.2. Least Gelation Concentration

The stink bean flour at different concentrations (2–20 g/100 mL) in dispersions was used to examine gelation properties (Table 4). SBRF exhibited partial and complete gelation at 10 g/100 mL and 14 g/100 mL, respectively. Notably, SBGF observed complete gelation at a concentration of 10 g/100 mL, suggesting that the germination process led to a decrease in the gelation concentration. These results align with prior research conducted on kidney beans and Nitta beans, which reported LGC at levels of 10 g/100 mL [45] and 12 g/100 mL [1], respectively. This could indicate that the amylase interacts with the starch molecules in the flour during germination, thereby augmenting its gelation characteristics. The noted differences in the gelation properties of flour can be attributed to the diverse ratios of lipids, carbohydrates, and proteins present in their composition. This suggests that the interplay among these components significantly influences the functional attributes [46]. SBHTCF achieved complete gelation at 12 g/100 mL, while SBR required 14 g/100 mL to form complete gelation.

#### 3.3.3. Emulsion, Swelling, Hydration, and Pasting Properties, Alongside Water and Oil Absorption Capacities

Table 5 presents the emulsion, swelling, and hydration characteristics of the flour, alongside its capacities for absorbing water and oil. The emulsion capacity and emulsion stability of SBHTC flour were significantly lower (*p* < 0.05) when compared to SBR and SBG flour. Similar findings were reported for soaked-cooked chickpeas [47]. However, a significant reduction in the EC and ES of SBHTC could be attributed to changes in the nature of its proteins induced by heat [48].

Emulsion stability in the bean flour is likely attributed to the presence of globular protein bodies. Notably, the EC and ES were higher in SBR and SBG, consistent with findings from yellow bean flour and faba bean flour [48]. The increased EC in SBR and SBG flour may be ascribed to the presence of more surface-active protein structures and increased surface hydrophobicity [31].

The SBG exhibited values of zero for both SC and SI. Furthermore, higher SC and SI were observed in SBR and SBHTC, possibly attributable to the elevated L* values of SBR and SBHTC. Nonetheless, the SC is influenced by the lightness of the bean, and this parameter can vary, potentially resulting in greater SC [49]. The size, shape, and thickness of the seed coat cover, as well as the cell structures and their compactness, are directly proportional to the HC of the seed [49]. Interestingly, the HC of SBHTC was significantly lower compared to that of SBR and SBG, implying that hydrothermal treatment alters the compactness of cells and cell structures, leading to starch and protein denaturation. This alternation may be a plausible explanation for the relatively lower HC of SBHTC.

The pasting characteristics, such as pasting temperature, peak viscosity, final viscosity, and peak time, were observed to be higher for SBG and SBHTC in comparison to SBR (Table 5). The pasting temperature corresponds to the lowest temperature required for cooking and the point at which the flour starts increasing viscosity rapidly [50]. The pasting temperature of SBG was notably higher than that of SBHTC and SBR. This increased pasting temperature of SBG and SBHTC may be attributed to interactions between starch, lipid, and protein complexes [51], which corroborates with the higher protein content in SBG and SBHTC compared to SBR. Peak viscosity reflects the strength of the paste resulting from gelatinization during processing [17]. Additionally, SBR exhibited significantly lower peak viscosity compared to SBG and SBHTC. Notably, no statistical significant disparity in peak viscosity was observed between SBG and SBHTC, consistent with findings reported for sprouted and cooked moth bean flour [18]. Final viscosity signifies a substance’s capacity to generate a viscous paste after cooking and subsequent cooling [52]. The trough and final viscosity of SBHTC flour were significantly higher in comparison to SBG and SBR. The lower trough and final viscosity values of SBG and SBR can be ascribed to reduced retrogradation, subsequently increasing the flour’s storage capacity [53]. These results are consistent with findings that demonstrate a decrease in the trough and final viscosity of pigeon pea, Dolichos beans, and jack beans after germination [54]. The minimum cooking time required for the flour is indicated by the peak time [54]. The peak time values for SBR, SBG, and SBHTC were lower than the peak time values reported by Acevedo et al. [54].

SBR demonstrated a significantly higher WAC compared to both SBG and SBHTC, indicating that prolonged germination leads to a reduction in WAC of germinated sorghum and brown rice flour [55,56]. The OAC of SBR, SBG, and SBHTC did not exhibit significant differences, aligning with findings in the OAC of Nitta bean raw and processed flour [1].

### 3.4. FTIR Analysis

Figure 1 depicts the spectra of FTIR for the flour obtained from SBR, SBG, and SBHTC across the spectral region spanning 500 to 4000 cm^−1^.

Within the examined spectral range (3100–3700 cm^−1^), a characteristic absorption band was observed, attributed to the vibrational stretching of OH groups in SBR, SBG, and SBHTC. An increase in absorbance was documented for SBG and SBHTC at 3300 and 3289 cm^−1^, respectively. Likewise, the spectral range of 3050–2800 cm^−1^ indicated absorption peaks, signifying the existence of vibrational patterns arising from stretching both asymmetric and symmetric CH bonds in triglycerides (lipid compounds) within the samples of stink beans.

The spectra revealed that there were no substantial differences in the absorbance of CH stretching between SBG and SBHTC when compared to SBR. Furthermore, the spectral range from 1750 to 1550 cm^−1^ exhibited the presence of the amide I and amide II bands associated with the proteins in the samples [32]. Amide I bands are generated by the stretching vibration of the C=O groups in the peptide structure [57], while amide II bands are attributed to the CO-NH stretching motion [33]. The absorbance spectra revealed an increased protein content in SBG and SBHTC compared to SBR, corroborating findings from studies on bean, soybean, chick peas [33], and Nitta bean [1]. The FTIR findings provided a comparative overview of the proximate composition for stink beans, showing a substantial increase in protein content in the processed stink bean.

The presence of carbohydrates in SBR, SBG, and SBHTC is indicated by the lower frequency bands within the range of 1200–900 cm^−1^ [33,58]. The spectral peak at 1075 cm^−1^, corresponding to the vibrational resonance of C-O groups within carbohydrates, exhibited heightened intensity in SBG. The FTIR results support the proximate composition of SBG, suggesting that germination reduces the overall carbohydrate content of stink beans. These findings align with previously reported studies [1,33].

### 3.5. Gelatinization Properties

The gelatinization properties of SBR, SBG, and SBHTC flour, including onset temperature (T0), peak temperature (TP), conclusion temperature (TC), temperature range (TC-T0), and gelatinization enthalpy, are depicted in Figure 2. These gelatinization properties were significantly impacted by processing methods such as germination and hydrothermal treatment.

The substantial difference between T0 and TC may be attributed to the pronounced variability in the gelatinization properties of the raw and processed flour molecules [52]. Specifically, the T0, TP, and TC of SBG (36.7, 82.6, 115.6) were lower compared to those of SBR (39.4, 83.2, 129.6), signifying notably reduced gelatinization characteristics in SBG flour. This reduction in gelatinization characteristics during germination can be ascribed to the decreased fat content typically enveloping starch granules, rendering them more susceptible to swelling [44]. Additionally, the decline in the conclusion temperature may be linked to the disruption of interactions between starch and protein or lipid constituents as a consequence of the germination process [17].

### 3.6. FE-SEM Analysis

FESEM was employed to evaluate the morphological characteristics of flour obtained from SBR, SBG, and SBHTC. Figure 3A–C illustrates the distinct presence of seed cells and cell walls within the central region that separates the lamellae, which were clearly observed at each level of magnification (×500, ×1000, and ×2000) in the SBR flour. The proteins and lipids integrated within the structures of both soluble and insoluble dietary fiber were distinctly apparent in the case of SBR flour. This observation aligns with the findings in flour obtained from chickpeas and lupin beans [33], as well as Nitta bean [1].

The SBG flour exhibited subtle morphological variations (Figure 3D–F) when compared to SBR, attributable to the heightened presence of proteins and lipids resulting from the germination process. Germination induces alterations in both seed starch and protein content, concurrently enhancing hydrolytic activity. This process leads to cell rupture and the discernible presence of starch precipitates, likely attributed to heightened proteolytic activity [59]. The observed structural modifications in germinated stink bean flour are most plausibly attributed to the elevated protein content and reduced carbohydrate levels, consistent with findings reported for germinated Nitta bean [1]. Within the stable cellular matrix, constituted primarily of densely compacted materials, spherical and oval protein bodies were discernible. After germination, cells were densely packed, resulting in the formation of substantial inter-cellular gaps. The protein structure displayed a granular pattern with multiple cracks, a feature attributable to seed germination, and was scarcely perceptible in the case of SBR. These findings are consistent with the flour of raw Nitta bean [1].

The SBHTC flour exhibited morphological disparities (Figure 3G–I), which may be linked to increased levels of proteins and lipids. A greater number of round and elliptical particles were observed within the extracellular interstices of the fibers. The heightened presence of both soluble and insoluble dietary fibers contributed to a smoother structural appearance. This outcome aligns with the findings reported for hydrothermally cooked Nitta bean flour [1].

### 3.7. Antioxidant Activity and Total Phenolic Content

Antioxidant potential (DPPH and ORAC) and total polyphenol content of SBR, SBG, and SBHTC are illustrated in Table 6. The SBG and SBHTC observed significantly higher antioxidant activity in comparison to the SBR, consistent with prior studies on the germination of lentils [60] and Nitta beans [1]. Hydrothermally cooked stink bean flour resulted in a decrease in both the ORAC antioxidant activity and polyphenol content, a finding supported by a previous study on hydrothermally cooked Nitta beans [1]. The germination process facilitates the release of bound phenolic compounds by amylases, proteases, and other hydrolytic enzymes [61]. In the present study, the SBG exhibited significantly higher TPC content compared to the SBR and SBHTC. The increased TPC might be ascribed to the heightened levels of phenylalanine ammonia-lyase (PAL) enzyme observed during the germination process [1,62].

## 4. Conclusions

The physiochemical attributes of stink beans underwent substantial enhancement through processing, with particularly notable improvements observed during germination. Furthermore, SBG demonstrated a significantly elevated protein content and diminished fat content (35.61 and 20.29 g/100 g) compared to SBHTC (33.68 and 28.73 g/100 g) and SBR (31.40 and 23.18 g/100 g). Notably, the least gelation concentration underwent significant alterations due to germination and hydrothermal cooking, decreasing from 10–14% for SBRF to 6–10% for SBG and 8–12% for SBHTC. Additionally, techno-functional properties, including color intensity, least gelation concentration, emulsion capacity and stability, hydration capacity, and pasting properties, exhibited substantial improvements in SBG compared to SBHTC and SBR. FTIR analysis of SBG and SBHTC revealed structural modifications in lipid, protein, and carbohydrate molecules. SEM examination highlighted morphological changes in SBG and SBHTC when compared to SBR. Importantly, SBG showcased heightened antioxidant activity and total phenolic content compared to SBHTC and SBR. In conclusion, the hydrothermal cooking and germination of stink beans presents promising avenues for achieving favorable outcomes in terms of protein content, total phenolic compounds, and antioxidant activity, as well as pasting and gelatinization properties. Consequently, processed stink bean flour can be effectively incorporated into various product developments, underscoring its potential as a plant-based protein source for protein-rich breakfast bars and cookies.

## Figures and Tables

**Figure 1 foods-12-04480-f001:**
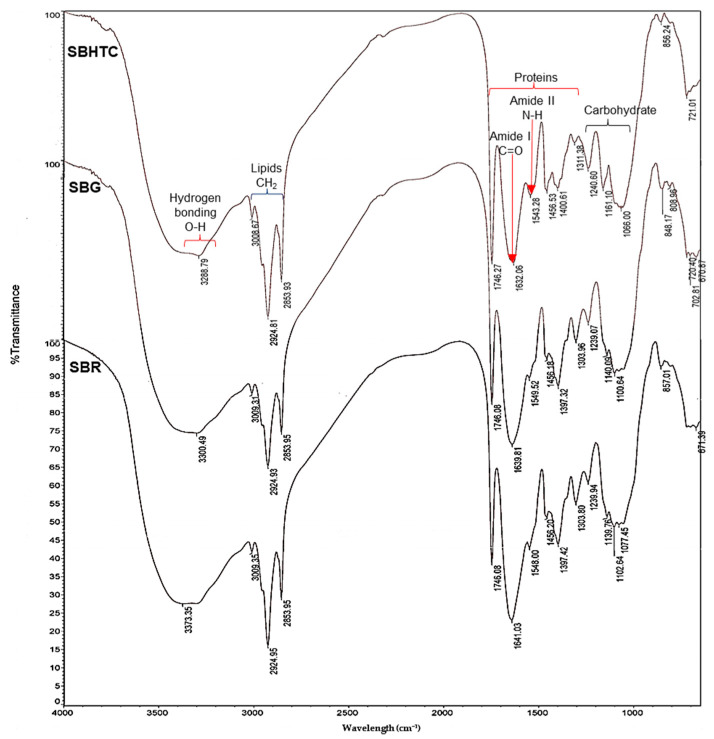
The spectra of FTIR for stink bean (SB) flour in their raw (R), germinated (G), and hydrothermally cooked (HTC) forms.

**Figure 2 foods-12-04480-f002:**
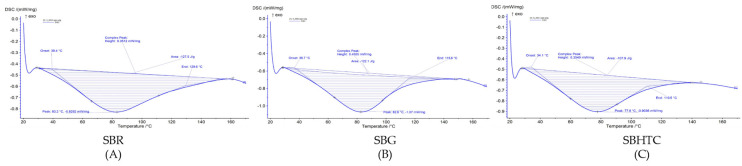
Gelatinization properties of stink bean (SB) flour in their raw (R) (**A**), germinated (G) (**B**), and hydrothermally cooked (HTC) (**C**) forms.

**Figure 3 foods-12-04480-f003:**
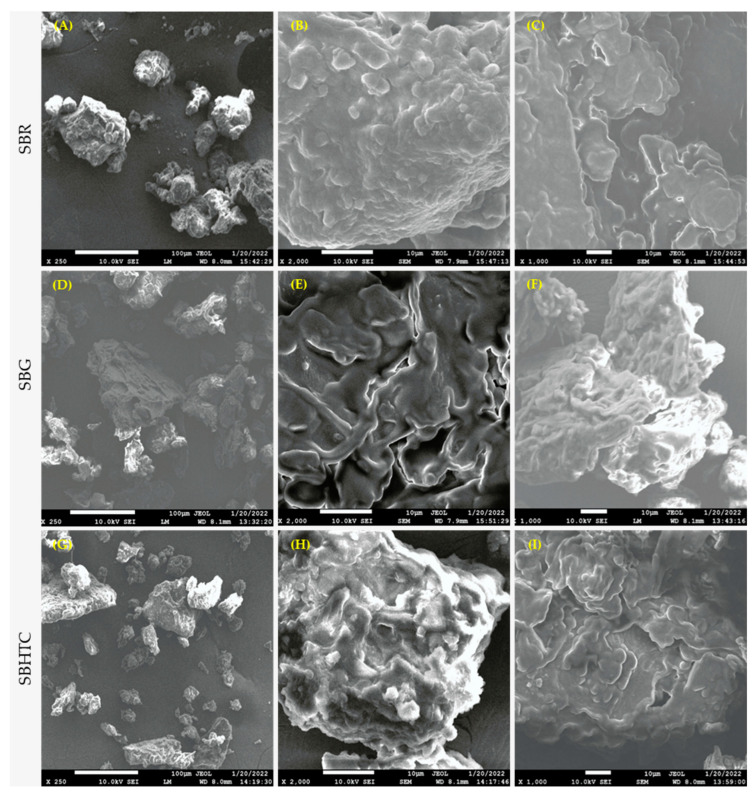
Morphology of stink bean (SB) flour in their raw (R) (**A**–**C**), germinated (G) (**D**–**F**), and hydrothermally cooked (HTC) (**G**–**I**) forms.

**Table 1 foods-12-04480-t001:** The physiochemical properties of stink bean (SB) seeds in their raw (R), germinated (G), and hydrothermally cooked (HTC) forms.

Parameter	SBR	SBG	SBHTC
Length (mm)	24.36 ± 0.22 ^b^	27.15 ± 0.24 ^c^	21.41 ± 0.21 ^a^
Width (mm)	17.97 ± 0.15 ^c^	17.95 ± 0.16 ^b^	15.45 ± 0.19 ^a^
Thickness (mm)	9.48 ± 0.10 ^b^	10.15 ± 0.16 ^c^	8.36 ± 0.11 ^a^
Diameter (mm)	16.05 ± 0.13 ^b^	16.68 ± 0.17 ^c^	14.00 ± 0.14 ^a^
Sphericity (%)	65.98 ± 0.28 ^b^	61.49 ± 0.37 ^a^	65.46 ± 0.30 ^b^
Aspect ratio	0.74 ± 0.00 ^b^	0.63 ± 0.01 ^a^	0.72 ± 0.01 ^b^
Volume (mm^3^)	1175.78 ± 27.98 ^b^	1323.20 ± 38.01 ^c^	787.56 ± 20.64 ^a^
Surface area (mm^2^)	685.74 ± 11.22 ^b^	742.97 ± 14.58 ^c^	523.97 ± 10.17 ^a^
Weight (g/100 seed)	197.33 ± 4.10 ^b^	271.40 ± 1.54 ^c^	154.28 ± 2.46 ^a^
Volume (mL/100 seed)	210.00 ± 2.89 ^b^	278.00 ± 3.06 ^c^	149.67 ± 0.33 ^a^

The results are presented as means ± standard error. Treatments are not significantly different (*p* > 0.05) when similar smaller superscripts are present within the same row.

**Table 2 foods-12-04480-t002:** Proximate composition of stink bean (SB) flour in their raw (R), germinated (G), and hydrothermally cooked (HTC) forms (g/100) on a dry weight basis.

Proximate Composition	SBR	SBG	SBHTC
Moisture	4.49 ± 0.13 ^B^	4.38 ± 0.02 ^B^	3.78 ± 0.07 ^A^
Total proteins	31.40 ± 0.50 ^A^	35.61 ± 0.06 ^C^	33.68 ± 0.04 ^B^
Total fats	23.18 ± 0.06 ^B^	20.29 ± 0.05 ^A^	28.73 ± 0.21 ^C^
Total carbohydrates *	39.16 ± 0.47 ^B^	37.97 ± 0.05 ^B^	33.56 ± 0.25 ^A^
Total dietary fibers	24.96 ± 0.09 ^B^	23.02 ± 0.05 ^A^	29.90 ± 0.09 ^C^
Soluble dietary fibers	7.06 ± 0.32 ^B^	5.26 ± 0.16 ^A^	8.52 ± 0.04 ^C^
Insoluble dietary fibers	17.90 ± 0.49 ^A^	17.76 ± 0.11 ^A^	21.38 ± 0.13 ^B^
Ash	6.26 ± 0.01 ^C^	6.14 ± 0.03 ^B^	4.03 ± 0.01 ^A^
Energy (kcal)	490.86 ± 0.60 ^B^	476.93 ± 0.17 ^A^	527.53 ± 0.57 ^C^

The results are presented as means ± standard error. Treatments are not significantly different (*p* > 0.05) when similar capital superscripts are present within the same row. * Difference used to calculate the carbohydrate values.

**Table 3 foods-12-04480-t003:** Hunter color parameters for raw (R), germinated (G), and hydrothermally cooked (HTC) stink bean (SB) seeds along with their respective flour (F).

		SBRF	SBGF	SBHTCF	SBR	SBG	SBHTC
Hunter color values	L	41.33 ± 0.04 ^a^	44.48 ± 0.03 ^b^	44.36 ± 0.01 ^b^	49.33 ± 0.15 ^B^	45.08 ± 0.12 ^A^	49.59 ± 0.17 ^B^
	a*	−4.92 ± 0.01 ^a^	−4.35 ± 0.01 ^b^	0.20 ± 0.01 ^c^	−9.21 ± 0.03 ^A^	−7.33 ± 0.03 ^B^	0.08 ± 0.03 ^C^
	b*	19.45 ± 0.05 ^a^	19.43 ± 0.02 ^a^	20.15 ± 0.00 ^b^	36.04 ± 0.07 ^C^	32.38 ± 0.02 ^B^	28.09 ± 0.15 ^A^
	Croma	20.07 ± 0.05 ^b^	19.91 ± 0.02 ^a^	20.15 ± 0.00 ^b^	37.20 ± 0.08 ^C^	33.20 ± 0.02 ^B^	28.09 ± 0.15 ^A^
	Hue Angle	104.20 ± 0.02 ^c^	102.62 ± 0.03 ^b^	89.44 ± 0.03 ^a^	104.34 ± 0.02 ^C^	102.76 ± 0.06 ^B^	89.84 ± 0.06 ^A^
Whitening Index		62.01 ± 0.03 ^c^	58.98 ± 0.03 ^a^	59.17 ± 0.01 ^b^	62.86 ± 0.08 ^B^	64.18 ± 0.11 ^C^	57.70 ± 0.22 ^A^
Browning Index		51.17 ± 0.13 ^b^	47.31 ± 0.08 ^a^	58.68 ± 0.03 ^c^	99.91 ± 0.21 ^C^	99.02 ± 0.49 ^B^	79.12 ± 0.97 ^A^

The results are presented as means ± standard error. Treatments are not significantly different (*p* > 0.05) when similar smaller and capital superscripts are present for flour and seeds within the same row.

**Table 4 foods-12-04480-t004:** Stink bean (SB) raw (R), germinated (G), and hydrothermally cooked (HTC) flour (F) with corresponding least gelation concentrations (g/100mL).

Concentration (g/100 mL)	SBRF	SBGF	SBHTCF
2	-	-	-
4	-	-	-
6	-	±	-
8	-	±	±
10	±	+	±
12	±	+	+
14	+	+	+
16	+	+	+
18	+	+	+
20	+	+	+

-: no gelation, ±: partial gelation, +: complete gelation.

**Table 5 foods-12-04480-t005:** The techno-functional properties of stink bean (SB) flour in their raw (R), germinated (G), and hydrothermally cooked (HTC) forms (g/100 g).

Parameter	SBR	SBG	SBHTC
Emulsion properties			
Capacity (%)	56.43 ± 2.49 ^a^	56.46 ± 1.19 ^a^	12.08 ± 1.13 ^b^
Stability (%)	56.10 ± 1.74 ^a^	55.49 ± 1.80 ^a^	3.80 ± 0.88 ^b^
Swelling properties			
Capacity (mL/seed)	0.91 ± 0.00 ^a^	0.00 ± 0.00	0.11 ± 0.03 ^a^
Index	0.01± 0.0 ^a^	0.00 ± 0.0	0.01± 0.0 ^a^
Hydration properties			
Capacity (g/seed)	0.51 ± 0.03 ^b^	0.48 ± 0.01 ^b^	0.11 ± 0.01 ^a^
Index	1.38 ± 0.03 ^a^	2.57 ± 0.03 ^b^	1.66 ± 0.11 ^a^
Pasting properties			
Temperature (°C)	55.85 ± 0.4 ^a^	86.95 ± 0.6 ^c^	74.5 ± 0.8 ^b^
Peak viscosity (cP)	2.16 ± 0.01 ^a^	4.96 ± 0.17 ^b^	4.70 ± 0.07 ^b^
Trough viscosity (cP)	0.46 ± 0.02 ^a^	0.44 ± 0.04 ^a^	2.79 ± 0.07 ^b^
Final viscosity (cP)	0.25 ± 0.01 ^a^	1.62 ± 0.07 ^b^	2.3 ± 0.12 ^c^
Peak time (min)	2.28 ± 0.01 ^a^	4.68 ± 0.08 ^b^	5.27 ± 0.03 ^c^
WAC (mL/g)	0.86 ± 0.0 ^a^	0.72 ± 0.00 ^b^	0.69 ± 0.03 ^b^
OAC (g/g)	0.86 ± 0.02 ^a^	0.81 ± 0.03 ^a^	0.83 ± 0.03 ^a^

The results are presented as means ± standard error. Treatments are not significantly different (*p* > 0.05) when similar smaller superscripts are present within the same row.

**Table 6 foods-12-04480-t006:** The antioxidant properties and total polyphenol content of stink bean (SB) flour in their raw (R), germinated (G), and hydrothermally cooked (HTC) forms.

Treatment	SBR	SBG	SBHTC
ORAC (mM TE/g)	862.2 ± 10.0 ^a^	1115.4 ± 68.1 ^b^	665.4 ± 22.9 ^c^
DPPH (mM TE/g)	44.8 ± 0.3 ^a^	59.8 ± 3.1 ^b^	66.0 ± 3.2 ^b^
TPC (mg GE/g)	6.05 ± 0.49 ^a^	9.25 ± 0.59 ^b^	3.40 ± 0.30 ^c^

The results are presented as means ± standard error. Treatments are not significantly different (*p* > 0.05) when similar smaller superscripts are present within the same row.

## Data Availability

Data is contained within the article.

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
