# Peer review of "Modification of Physiochemical and Techno-Functional Properties of Stink Bean (Parkia speciosa) by Germination and Hydrothermal Cooking Treatment"

_foods, 2023, doi:10.3390/foods12244480_

Round 1

Reviewer 1 Report

Comments and Suggestions for Authors

A very well-written introduction, on the one hand synthetic, but drawing attention to the most important factors from the point of view of the research undertaken.

in the first part of the methodology, chapters from 2.2.2 to 2.2.6. - there is no description of easy markings at all, and the lack of a legend for the formulas from which the marked quantities were calculated. Moreover, what is the point of including numbers in formulas if the authors do not refer to them in the text?

line 72 "2.1.1. Stink bean raw (NBR)" from the work results from the wrong abbreviation, shouldn't it be SBR?

tab 2 how the caloric value was determined, there is nothing about it in the methodology

line 428 - "SBHT could be due to the elevated L* values of SBR and SBHT" - what kind of sample is this, it cannot be mentioned in the text.

The authors put a lot of work into the marking planned in the work. I have the impression that they are not fully thought out, they are necessary. Some of the methodology was very carelessly developed. The authors confuse the abbreviations of samples in the work, which makes it very difficult to freely navigate through the text and its results. The part related to the generally defined size, etc. of beans was too extensive, not to mention the fact that it was not described reliably. The work requires thorough editing.

Author Response

Response to Reviewer 1 Comments

Point 1: A very well-written introduction, on the one hand synthetic, but drawing attention to the most important factors from the point of view of the research undertaken.

Response: Thank you for your appreciation and timely effort in reviewing our article.

Point 2: in the first part of the methodology, chapters from 2.2.2 to 2.2.6. - there is no description of easy markings at all, and the lack of a legend for the formulas from which the marked quantities were calculated. Moreover, what is the point of including numbers in formulas if the authors do not refer to them in the text?

Response: Thank you for the suggestion, and we sincerely apologize for the oversight in not including the description of the legend in methodology chapters 2.2.2. to 2.2.6. The necessary details have now been incorporated into the methodology section.

Moreover, what is the point of including numbers in formulas if the authors do not refer to them in the text?

The Foods Journal format requires numbering for each formula or equation, and we have accordingly incorporated numbered formulas into the manuscript.

In the revised manuscript

Lines 127-141:

2.2.2. Equivalent diameter

Seeds equivalent diameter (Dm) was evaluated using the formula below

Dm = (LWT)1/3

(1)

Where L = length, W = width, T = thickness

2.2.3. Sphericity

The calculation of sphericity (ɸ) was performed using the formula below

= [(LWT)1/3/L] × 100

(2)

Where L = length, W = width, T = thickness

2.2.4. Aspect ratio

The seeds aspect ratio (Ra) was determined using the following formula

Ra = W/L

(3)

Where W = width, L = length

2.2.5. Seed volume

The seed volume (V) was determined using the below-given formula

V = πB2L2/6(2L-3)

(4)

Where, B = (WT)1/2, L = length, W = width, T = thickness

2.2.6. Surface area

The surface area (A) was determined using the following formula

A = πBL2/2L-B

(5)

Where, B = (WT)1/2, L = length

Point 3: line 72 "2.1.1. Stink bean raw (NBR)" from the work results from the wrong abbreviation, shouldn't it be SBR?

Response: Thank you for the excellent observation, and we sincerely apologize for the typing error. The necessary details are now included in the methodology section, specifically in line 89.

Point 4: tab 2 how the caloric value was determined, there is nothing about it in the methodology

Response: Thank you for the excellent observation, and we sincerely apologize for not including the formula to determine the caloric value. The caloric value was calculated based on the reference value provided by FAO (2003). We have now included the necessary details in the revised manuscript.

Reference

FAO. Chapter 3: Calculation of the energy content of foods-Energy conversion factors. In Proceedings of the Report of a Technical Workshop. Paper 77, 2003.

In the revised manuscript

Lines 162-163:

The energy (kcal/100 g) was assessed using the below-given equation [22],

Energy = (%Carbohydrate × 4 kcal) + (%Protein × 4 kcal) + (%Fat × 9 kcal)

(6)

Point 5: line 428 - "SBHT could be due to the elevated L* values of SBR and SBHT" - what kind of sample is this, it cannot be mentioned in the text.

Response: Thank you for the excellent observation, and we sincerely apologize for the typing error. We have now included the necessary details in the revised manuscript.

In the revised manuscript

Lines 432-434:  Furthermore, higher SC and SI were observed in SBR and SBHTC, possibly attributable to the elevated L* values of SBR and SBHTC.

Point 6: The authors put a lot of work into the marking planned in the work. I have the impression that they are not fully thought out, they are necessary. Some of the methodology was very carelessly developed. The authors confuse the abbreviations of samples in the work, which makes it very difficult to freely navigate through the text and its results. The part related to the generally defined size, etc. of beans was too extensive, not to mention the fact that it was not described reliably. The work requires thorough editing.

Response: Thank you for your observation. We acknowledge your concerns regarding the marking and methodology of our work. Recognizing the need for a more detailed explanation and thoughtful development, we have provided a comprehensive and well-thought-out methodology in our revised manuscript to ensure clarity and accuracy.

We apologize for any confusion related to the abbreviations of samples in the manuscript. In the revised version, we have introduced a clear and consistent set of abbreviations, enhancing the readability and navigation of the text.

We also understand your point about the extensive description of the size of beans. As our focus is on determining the physiochemical properties of raw and processed beans, it was important to assess the physical parameters of stink beans. This information is crucial for various aspects, including processing, postharvest handling, transport, and developing equipment for postharvest treatments.

Reviewer 2 Report

Comments and Suggestions for Authors

In this study, the effects of germination (G) and hydrothermal cooking (HTC) on the physicochemical properties, proximate composition, process functional properties and antioxidant activity of stinky bean (SB) flour were investigated. Overall, the paper has certain novelty and advantages for this field research work, and has value. I suggest this manuscript needs the following revisions:

1. Abstract must have rationale, objective, materials and methods and conclusions.

 2. Please further distill the introduction, rendering the research background exposition more lucid. According to the actual processing and use of the sample, the importance of sprouting and cooking methods to the use of the sample is further explained.

3. All data must be presented with appropriate decimal point based on the standard deviation. For examples:78.99+8.67 should be 79+9ï¼›78.99+0.67 should be 79.0+0.7ï¼›78.99+0.073 should be 78.99+0.07

4. Figure 1 needs to be standardized for graphing, is the vertical scale labeling correct? If multiple spectra are superimposed, it is recommended that the vertical scale labeling be removed. Peak positions need to be labeled with a larger font size for clarity. In addition, the labeling of peaks that are not relevant to the experimental results can be omitted.

5. It is suggested to add the data and results of the highlights of this experiment in the conclusion section to increase the persuasiveness of the conclusion

Author Response

Response to Reviewer 2 Comments

Point 1: In this study, the effects of germination (G) and hydrothermal cooking (HTC) on the physicochemical properties, proximate composition, process functional properties and antioxidant activity of stinky bean (SB) flour were investigated. Overall, the paper has certain novelty and advantages for this field research work, and has value.

Response: Thank you for your appreciation and timely effort in reviewing our article.

I suggest this manuscript needs the following revisions:

Point 2: Abstract must have rationale, objective, materials and methods and conclusions.

Response: Thank you for the suggestion. We have included rationale in the abstract and objective, materials and methods, and conclusion, which was already included in the previous version of the manuscript.  The rationale, objective, materials and, methods and conclusions of the abstract are included in the below-given information

Rationale: Stink bean, Parkia speciosa, is recognized as a significantly underutilized legume with versatile utility and diverse benefits. However, information on the impact of different processing methods, such as germination and hydrothermal cooking, is scarce on stink beans (SBs).

Objective: Therefore, the current research aimed to explore the efficacy of germination (G) and hydrothermal cooking (HTC) on the physiochemical properties, proximate composition, techno-functional properties, and antioxidant potential of SB flour.

Materials and Methods: Physiochemical properties, proximate composition, techno-functional properties, and antioxidant potential, Fourier transform infrared spectroscopy (FTIR) and field emission scanning electron microscopy (FESEM)

Results: The results revealed that the physiochemical properties of SB were significantly enhanced through processing, with more pronounced improvements observed during germination. Additionally, SBG exhibited a significantly higher protein content and lower fat content compared to SBHTC and stink bean raw (SBR). Moreover, techno-functional properties such as color intensity, least gelation concentration, and pasting properties were significantly improved in SBG compared to SBHTC and SBR. FTIR analysis of SBG and SBHTC indicated structural modifications in the lipid, protein, and carbohydrate molecules. FESEM examination revealed morphological changes in SBG and SBHTC when compared to SBR. Importantly, SBG exhibited higher antioxidant activity and total phenolic content in comparison to SBHTC and SBR.

Conclusion: Therefore, processed SB flour can be incorporated and utilized in product development, highlighting its potential as plant-based protein source for protein-rich breakfast bars and cookies.

In the revised manuscript

Lines 14-16:  Stink bean, Parkia speciosa, is recognized as a significantly underutilized legume with versatile utility and diverse benefits. However, information on the impact of different processing methods, such as germination and hydrothermal cooking, is scarce on stink beans (SBs).

Point 3: Please further distill the introduction, rendering the research background exposition more lucid. According to the actual processing and use of the sample, the importance of sprouting and cooking methods to the use of the sample is further explained.

Response: Thank you for the constructive comments aimed at improving the quality of the paper. We found your suggestions extremely helpful and have revised the manuscript accordingly. All the recommendations have been incorporated into the introduction of the revised manuscript.

In the revised manuscript

Lines 66-80:  Basic household methods like cooking and germination have emerged as a budget-friendly strategy for addressing anti-nutritional factors in beans, enhancing their sensory appeal (Wani et al., 2013). Germination, a controllable and straightforward method, holds promise in augmenting the nutritional profile of grains. Under controlled humidity, grains germinate after a specific soaking period in water. This process leads to the breakdown of key nutrients like proteins, starch, lipids, and fibers, accompanied by increased hydrolytic enzyme activity. This transformation, in turn, reduces the cooking time of hard-to-cook seeds and enhances various functional aspects (Li et al., 2020). On the other hand, cooking triggers starch gelatinization, protein denaturation, and the solubilization of polysaccharides. The increasing desire for innovative raw materials in the functional food industry has driven a consistent market demand for beans processed using techniques such as germination and hydrothermal cooking. These techniques have proved effective in improving both the functional and nutritional qualities of beans (Medhe et al., 2022; Medhe et al., 2019). Based on our current knowledge, there is a notable absence of published data on the physiochemical characteristics of raw, germinated, and hydrothermally cooked stink bean flour.

References

Li, C.; Jeong, D.; Lee, J.H.; Chung, H.-J. Influence of germination on physicochemical properties of flours from brown rice, oat, sorghum, and millet. Food Science and Biotechnology 2020, 29, 1223-1231.

Medhe, S.; Jain, S.; Anal, A.K. Effects of sprouting and cooking processes on physicochemical and functional properties of moth bean (Vigna aconitifolia) seed and flour. Journal of Food Science and Technology 2019, 56, 2115-2125.

Medhe, S.V.; Kamble, M.T.; Kettawan, A.K.; Monboonpitak, N.; Kettawan, A. Effect of hydrothermal cooking and germination treatment on functional and physicochemical properties of Parkia timoriana bean flours: An underexplored legume Species of Parkia genera. Foods 2022, 11, 1822.

Wani, I.A.; Sogi, D.S.; Gill, B.S. Physicochemical and functional properties of flours from three Black gram (Phaseolus mungo L.) cultivars. International Journal of Food Science & Technology 2013, 48, 771-777.

Point 4: All data must be presented with appropriate decimal point based on the standard deviation. For examples:78.99+8.67 should be 79+9ï¼›78.99+0.67 should be 79.0+0.7ï¼›78.99+0.073 should be 78.99+0.07

Response: Thank you for the suggestion. We aim to present our data with two decimal places in means ± standard error. With due respect to the reviewer, to uphold the precision and reliability of the data in this paper, we have meticulously ensured that all numerical values maintain consistency by adhering to a two-decimal-point precision throughout the document. Our previously published article also followed the two-decimal-point with means ± standard error (Medhe et al., 2022).

References

Medhe, S.V.; Kamble, M.T.; Kettawan, A.K.; Monboonpitak, N.; Kettawan, A. Effect of hydrothermal cooking and germination treatment on functional and physicochemical properties of Parkia timoriana bean flours: An underexplored legume Species of Parkia genera. Foods 2022, 11, 1822.

Point 5: Figure 1 needs to be standardized for graphing, is the vertical scale labeling correct? If multiple spectra are superimposed, it is recommended that the vertical scale labeling be removed. Peak positions need to be labeled with a larger font size for clarity. In addition, the labeling of peaks that are not relevant to the experimental results can be omitted.

Response: Thank you for the suggestion. We have followed the reviewer's advice to remove the vertical scale labeling. The peak positions were generated from the software used in FTIR analysis. To enhance clarity, we have increased the contrast. The revised Figure 1 has been included in the revised manuscript.

In the revised manuscript

Lines 470-472: 

Figure 1. The spectra of FTIR for stink bean (SB) flours in their raw (R), germinated (G), and hydrothermally cooked (HTC) forms.

Point 6: It is suggested to add the data and results of the highlights of this experiment in the conclusion section to increase the persuasiveness of the conclusion

Response: Thank you for the suggestion to improve the quality of the article. The data and results highlighting key aspects of this experiment have been included in the conclusion of the revised manuscript.

In the revised manuscript

Lines 562-580: The physiochemical attributes of stink beans underwent substantial enhancement through processing, with particularly notable improvements observed during germination. Furthermore, SBG demonstrated a significantly elevated protein content and diminished fat content (35.61 and 20.29 g/100g) compared to SBHTC (33.68 and 28.73 g/100g) and SBR (31.40 and 23.18 g/100g). Notably, the least gelation concentration underwent significant alterations due to germination and hydrothermal cooking, decreasing from 10-14% for SBRF to 6-10% for SBG and 8-12% for SBHTC. Additionally, techno-functional properties, including color intensity, least gelation concentration, emulsion capacity and stability, hydration capacity, and pasting properties, exhibited substantial improvements in SBG compared to SBHTC and SBR. FTIR analysis of SBG and SBHTC revealed structural modifications in lipid, protein, and carbohydrate molecules. SEM examination highlighted morphological changes in SBG and SBHTC when compared to SBR. Importantly, SBG showcased heightened antioxidant activity and total phenolic content compared to SBHTC and SBR. In conclusion, hydrothermal cooking and germination of stink beans present promising avenues for achieving favorable outcomes in terms of protein content, total phenolic compounds, antioxidant activity, as well as pasting and gelatinization properties. Consequently, processed stink bean flour can be effectively incorporated into various product developments, underscoring its potential as a plant-based protein source for protein-rich breakfast bars and cookies.

Round 2

Reviewer 1 Report

Comments and Suggestions for Authors

I see significant improvement in the way results are presented and discussed. My comments were taken into account and errors were corrected

Reviewer 2 Report

Comments and Suggestions for Authors

The authors have made sufficient modifications according to the modification comments. Overall, the data of this paper is relatively substantial and the analysis is proper. The manuscript in its present version is apposite for publication in Foods, and I suggest that this paper be accepted without further modification.